# Dietary Ground Flaxseed Increases Serum Alpha-Linolenic Acid Concentrations in Adult Cats

**DOI:** 10.3390/ani12192543

**Published:** 2022-09-23

**Authors:** Matthew R. Panasevich, Leighann Daristotle, Ryan M. Yamka, Nolan Z. Frantz

**Affiliations:** 1Blue Buffalo Co., Ltd., 11 River Rd., Wilton, CT 068797, USA; 2Luna Science & Nutrition LLC, Trumbull, CT 06611, USA

**Keywords:** cats, flaxseed, alpha-linolenic acid, bioavailability, omega-3 fatty acids

## Abstract

**Simple Summary:**

The use of flaxseed in complete pet foods as a source of omega-3 fatty acids, and in particular alpha-linolenic acid, has been studied extensively in dogs. However, to date, no scientific published literature exists on the impacts of feeding whole ground flaxseed as an ingredient in cat food on nutrient digestibility, fatty acid bioavailability, and overall health in cats. In addition, dietary flaxseed inclusion has been challenged as providing little nutritional value for cats by not changing serum fatty acid levels and lowering nutrient digestibility. Therefore, the objective of this study was to directly evaluate the effects of adding ground flaxseed to dry cat food on fatty acid bioavailability, stool consistency, nutrient digestibility, and overall health in adult cats. These results clearly exhibit that a flaxseed-supplemented diet resulted in no changes in stool quality or nutrient digestibility, and elicited an increase in serum alpha-linolenic acid concentrations in healthy cats. Overall, these results clearly demonstrate that ground flaxseed can be used to modulate serum fatty acid concentrations and serve as a bioavailable source of alpha-linolenic acid in commercial cat food.

**Abstract:**

We evaluated effects of dietary ground flaxseed on fecal and serum alpha-linolenic acid (ALA) concentrations, nutrient digestibility, and stool quality in female and male adult cats (n = 20 (8 males, 12 females); 3.95 ± 1.49 years of age (mean ± SD); 3.88 ± 0.82 kg BW). We hypothesized that adding ground flaxseed would increase serum ALA compared with feeding no flax, without changing nutrient digestibility. Cats were fed as-is 2.6% added-flaxseed (flax, n = 10) or no-flax (control, n = 10) diets (2.66 vs. 0.78% ALA of total fatty acids; crude protein 35%, fat 20%, fiber 3% as-fed) twice daily to maintain body weight for 28 days. Fecal collections were conducted on days 23–27 for total-tract nutrient digestibility, stool quality (scale 1–5; 1 = watery diarrhea, 5 = hard, dry, crumbly) and long-chain fatty acid (LCFA) analyses. Blood was collected on days 0, 14, and 28 for serum LCFA and chemistry analysis. Digestibility and fecal data were analyzed by ANOVA (SAS v9.4, Cary, NC, USA) and a repeated measures ANOVA for serum ALA. Flax-fed cats, compared with control-fed, had greater (*p* < 0.05) serum ALA after 14 days (4.00 vs. 0.71 µg/mL) and 28 days (7.83 and 3.67 µg/mL). No differences were observed in stool quality, and dry matter, protein, fat, and ALA digestibility. However, metabolizable energy was greater in the flax vs. control diet (4.18 vs. 3.91 kcal/g; *p* < 0.05). Overall, these data demonstrate that ground flaxseed added to cat diets increases serum ALA within 14 days, with no detriments to nutrient digestibility. We conclude that flaxseed can be used as a bioavailable source of ALA in cat diets.

## 1. Introduction

Pet foods that provide nutritional benefits beyond the basic daily requirements for dogs and cats are in constant demand by pet owners. Crude fat, omega-3 and 6 fatty acids, eicosapentaenoic acid (EPA; C20:5n-3), and docosahexaenoic acid (DHA; C22:6n-3) are commonly expressed on pet food labels in the guaranteed analysis, as they provide numerous benefits to animals as well as supply required energy. Fatty acids that must be derived from the diet due to an inability of human or animal de novo synthesis are essential fatty acids. Specifically, polyunsaturated omega-3 fatty acids (n-3 PUFA), and in particular alpha-linolenic acid (ALA; C18:3n-3), and omega-6 linoleic acid (LA; C18:2n-6) are notable essential fatty acids. Eicosapentaenoic acid, docosapentaenoic acid (DPA; C20:4n-3), and to some degree DHA are synthesized from ALA in most mammalian species. Other long-chain fatty acids (LCFA), and especially essential fatty acid arachidonic acid (ARA; C20:4n-6), are synthesized from LA in most mammals, with cats being the notable exception.

Flaxseed contains a substantial concentration of essential fatty acids ALA (23%) and LA (6.5%), as well as protein (20%), fiber (28%), and antioxidant compounds (e.g., lignans) [1,2]. Many investigations of feeding flaxseed and flaxseed oil have been evaluated in dogs. Specifically, adult dogs fed an ALA-rich flaxseed for 84 days exhibited higher serum concentrations of ALA, DPA, and EPA, but not DHA, just 4 days after feeding, which is a finding consistent with other mammalian species [3]. Flaxseed oil has previously been investigated in greyhounds and found to significantly increase plasma ALA concentrations [4] and result in improvements in inflammatory genes that were negatively correlated to ALA and EPA [5]. Furthermore, ALA and LA increased from d 15 and d 22, suggesting that PUFA are not stabilized after 3 weeks of feeding [5]. Interestingly, in dogs with atopic dermatitis, supplementation with flaxseed oil for 10 weeks resulted in improved clinical scores; however, the omega-6:omega-3 PUFA ratio was not correlated with improvements in clinical scores [6].

In 2017, the Association of American Feed Control Officials (AAFCO) updated the nutrient requirements for cats to include LA at 1.40 g per 1000 kcal ME for maintenance, growth and reproduction and ALA at 0.05 g per 1000 kcal ME for growth and reproduction, to be consistent with the 2006 National Research Council RA and 2007 AAFCO Feline Nutrition Expert Subcommittee guidelines [7,8]. It was also stated that no requirement for ALA in adult cats could be made [8]. Given that the activity of delta-6 desaturase enzyme is presumably low in cats [9] and therefore cannot effectively convert ALA to DPA and EPA, it is often viewed that only fat sources such as fish oil that are high in DHA and EPA can deliver health benefits in cats.

However, the mechanistic insight into ALA’s anti-inflammatory impact has been demonstrated in vitro by modulating NF-κB and mitogen-activated protein kinase [10,11]. More recently, ALA has been linked to decreasing pro-inflammatory mediators from T-cells, suggesting a direct effect of ALA on suppressing inflammation [12]. Furthermore, it has been observed that feeding flaxseed oil for 12 weeks to adult cats resulted in increased serum concentrations of ALA [13]. Specifically, 12 weeks of feeding a diet containing flaxseed oil reduced skin inflammatory responses to histamine and lowered B-cell, total T-cell, T-helper subset populations, and leukocyte proliferation in response to pokeweed mitogen in cats [13]. To date, no studies have investigated whole ground flaxseed inclusion in feline diets and its impacts on fatty acid concentrations, and in particular, ALA. Typically oil rather than the ground seed elicits greater bioavailability of ALA [14]. Furthermore, whole ground flaxseed has been challenged as being indigestible and impacting ALA bioavailability due to the appreciable levels of dietary fiber and lignans, although there are no scientific peer-reviewed studies to support this viewpoint. It is often misunderstood that flaxseed in pet food formulations is whole-non ground seeds that would result in no impact on serum fatty acid concentrations. However, most pet food formulations utilize ground whole flaxseed. 

In this study, our objective was to evaluate the bioavailability of n-3 PUFAs, specifically ALA, by feeding diets with and without the addition of ground flaxseed in adult cats. We hypothesized that dietary ground flaxseed would elicit an increase in serum ALA concentrations with minimal influence on nutrient digestibility and stool consistency and be a viable source of modulating omega-3 fatty acid concentrations in finished product formulations.

## 2. Materials and Methods

### 2.1. Animals and Diets

Twenty (12 female; ten spayed and two intact and 8 male; one intact and seven neutered) clinically normal adult cats (age: 3.95 ± 1.49 years of age; mean ± standard deviation; body weight: 3.88 ± 0.82 kg) were studied in a 28-d, 2-dietary treatment, longitudinal experiment. The protocol was first approved by the facility’s Institutional Animal Care and Use Committee (Summit Ridge Farms; Susquehanna, PA, USA) prior to feeding. Physical exams, as well as serum chemistry results for normal physiologic ranges before, during, and after the study were used to confirm health of all animals. Housing for all cats was individual, temperature-regulated, and equal light-dark cycles (12 h each). Weekly individual body weights were used to determine adjustments in daily food offerings (fed once per day) to target body weight maintenance. Food offerings remained constant during the digestibility collections (end of study, d 23–27). Before beginning the study, all animals were fed an adult maintenance cat diet for at least 2 weeks that utilized beef tallow as the fat source, which is void of ALA.

Ingredient inclusion and analyzed nutrient composition of the experimental diets is shown in Table 1. Formulation of all experimental diets was targeted to meet the AAFCO requirements for adult maintenance. Flaxseed was added to the diet at 2.6% as-is at the expense of brown rice, oats, and barley in a dry extruded kibble diet.

### 2.2. Fecal Collection, Consistency Scoring, and Blood Collection

Total fecal collections were done on the last 5 days of the study for nutrient digestibility and fecal scoring. Fecal quality was assessed three times per day, utilizing a 5-point scale (0 = none; 1 = watery diarrhea; 1.5 = diarrhea; 2 = moist, no form; 2.5 = moist, some form; 3 = moist, formed 3.5 = well-formed, sticky 4 = well-formed; 4.5 = hard, dry; and 5 = hard, dry, crumbly) during the 5-d digestibility period [15].

Approximately a 5-mL blood sample from each cat was taken via jugular venipuncture at days 0, 14, and 28 for LCFA, hematology, and chemistry profiles. Blood samples were allocated to either serum tubes and centrifuged for 15 min at 3000× *g* rpm after clotted, or tubes containing ethylenediaminetetraacetic acid anticoagulant. Remaining serum for LCFA analysis was allocated to 1.5-mL cryovial tubes and frozen at −80 °C until analysis. 

### 2.3. Total Tract Apparent Nutrient Digestibility 

Total tract apparent nutrient digestibility fecal collections were done at the end of study (d 23–27). All fecal and diet samples were analyzed at a contract laboratory (Eurofins US, Des Moines, IA, USA) according to the Association of Offical Analytical Chemists (AOAC) methods for moisture, fat, crude protein, and energy (AOAC 930.15, AOAC 954.02, AOAC 990.03, AOAC 962.0, AOAC 992.15, AOAC 991.43). 

Calculations for nutrient and energy digestibility were done as follows: Nutrient Digestibility (%) = [intake of nutrient (g/d) − fecal output (g/d)]/intake of nutrient (g/d) × 100%.

### 2.4. Fecal and Serum Fatty Acid and Chemistry Analyses

To identify ALA fecal and serum concentrations, total fatty acids were quantified according to Hahn et al. [16]. Briefly, fecal and serum samples were analyzed by gas chromatography. Samples (~100 mg or 100 µL) were subjected to a one-step, direct transesterification procedure to generate fatty acid methyl esters [17]. Fecal samples were lyophilized prior to transesterification as a precaution to prevent reaction interference from water. Two internal fatty acid standards (11:0 and 23:0) were used within the assay, and a quantitative external standard was used to measure response factors for the flame ionization detector. Fatty acids of interest were identified and quantified by comparing retention times with known fatty acids contained in a mixed standard (Supelco 37 Component FAME Mix; Supelco, Bellefonte, PA, USA). A sample peak was defined as being 3-times the height of background noise. Peak less than 3-times the background noise were considered not detectable.

Quantification of fatty acid methyl esters was performed on gas chromatographers (either model 6890, Agilent, Santa Clara, CA or model 5890 Series II, Hewlett-Packard, Palo Alto, CA, USA) equipped with a 100 m × 0.25 mm i.d. × 0.2 µm film thickness capillary column (model SP-2560; Supelco, Bellefonte, PA, USA). Injection (25:1 split ratio) was performed at 240 °C and detection was performed at 245 °C. Starting column head pressure was 48 psi with a constant helium flow of 1.6 mL/min. The oven temperature program was as follows: 120 °C isothermal for 7 min, increased to 180 °C at 1 °C/min and held for 8 min, increase to 240 °C at 1 °C/min and held for 12 min, decreased to 120 °C at 20 °C/min and held for 9 min. Antech Diagnostics (Memphis, TN, USA) analyzed blood samples for hematology (Siemens Advia 120) and serum clinical chemistries (Beckman Coulter AU5800).

### 2.5. Statistical Analysis

Between-diet effects for apparent total-tract nutrient digestibility, average weekly food intakes, final BW, fecal score, total fecal output, total fecal fatty acids, and fecal ALA were analyzed using Mixed models ANOVA in SAS (version 9.4; SAS Institute, Cary, NC, USA), where the fixed effect was diet and the random effect was cat. A repeated measures 2-way ANOVA was used for serum LCFA concentrations over time. Data are presented as mean and pooled SEM or individual timepoint SEM for serum ALA, hematology, and serum chemistry results. Statistical significance was set at *p <* 0.05. 

## 3. Results

Average weekly food consumption, final BW, and nutrient digestibility data are presented in Table 2. No differences were observed between dietary treatments for body weight and food intake. No food refusals were noted throughout the study. No differences in dry matter, crude protein, crude fat, energy, and ALA total-tract nutrient digestibility were observed between dietary treatments; however, lower (*p <* 0.05) metabolizable energy was observedin control versus flaxtreatment. Table 3 exhibits fecal score, total fecal output, total fecal fatty acids, and fecal ALA concentrations. No differences were observed in fecal score, total fecal fatty acid concentrations, and fecal ALA concentrations. Total fecal output was greater (*p <* 0.05) in flax-fed compared with control-fed cats. Figure 1 shows the serum ALA concentrations at days 0, 14, and 28 of feeding control and flax diets to adult cats. Serum ALA concentrations were greater (*p <* 0.05) at days 14 and 28 in cats fed flaxseed compared with cats fed control diet.

Hematology and serum chemistry values were within normal healthy reference range for all cats (Table 4). Cats consuming flaxseed exhibited greater (*p <* 0.05) serum potassium and platelets, while control-fed cats had lower (*p <* 0.05) serum mean corpuscular hemoglobin concentration, compared with flaxseed-supplemented cats. 

## 4. Discussion

Providing sources of omega-3 and -6 fatty acids through ingredients is commonly found in pet food formulations. Formulation strategies that target optimal fatty acid balance of omega-3: omega-6 fatty acids, and providing sources of EPA and DHA are critical for mitigating age-related diseases (e.g., obesity, osteoarthritis, and renal disease) [18,19,20]. Common ingredients such as fish oil and canola oil are utilized to modulate omega-3 PUFA in relation to omega-6 fatty acids. Flaxseed is a common pet food ingredient in complete and balanced formulas as well as supplements and treats due to its favorable fatty acid content (omega-3 and -6 fatty acids), protein, and fiber fractions. Specifically, flaxseed has a preponderance of ALA, which in dogs and most mammalian species can be converted to DPA, EPA, and to some extent DHA, which are the focus of health benefits and are commonly found in fish ingredients. However, the use of flaxseed as a source of omega-3 fatty acids, and in particular the fatty acid ALA, in cat diets is often dismissed due to their inability to convert ALA to EPA and DHA. More recent evidence in human, companion animal, and pre-clinical models suggests that ALA itself has immunomodulatory benefits [5,13,21,22]. Although it has been reported that cats have low delta-6 desaturase activity, there is evidence that at least in liver and brain tissue specifically, synthesis of long-chain omega-3 fatty acids can be achieved in the absence of dietary EPA, DPA, and DHA in cats [23]. Being that whole ground flaxseed is high in fiber and lignans, it is often criticized as being unable to be a source of bioactive fatty acid sources such as ALA. In this study, our objective was to evaluate whether addition of whole ground flaxseed can increase ALA concentrations without sacrificing apparent total-tract nutrient digestibility. We hypothesized that inclusion of flaxseed would result in negligible differences in nutrient digestibility and increase ALA concentrations in healthy adult cats. To our knowledge, this is the first study investigating whole ground flaxseed inclusion in a cat diet and its impacts on serum ALA.

We found no differences in dry matter, crude protein, crude fat, energy, or apparent total-tract nutrient digestibility. The chemical composition of flaxseed shows that it is favorable in total dietary fiber (32%), with a preponderance of insoluble fiber (23.7%) compared with soluble fiber (8.9%) [24]. As assessed by organic matter disappearance over a 24 h in vitro fermentation assay, flaxseed, apple pomace, and carrot pomace exhibited the greatest fermentability; however, flaxseed exhibited low amounts of gas and short-chain fatty acid production [24]. Higher inclusions of moderate to low fermentable fiber can decrease total-tract nutrient digestibility [25]. Indeed, dietary supplementation of flaxseed has been observed to decrease apparent nutrient digestibility in dogs and humans [26,27]. Oil-removed flaxseed (linseed cake) fed up to 8% in extruded diets fed to adult working Alaskan huskies resulted in linear decreases in OM, fat, neutral detergent fiber, nitrogen-free extract, and crude carbohydrates digestibility, and increased fecal dry matter and total wet fecal output [28]. It was concluded from that study that inclusion up to 4.2% can be well tolerated and exhibit minimal detriments on total-tract nutrient digestibility. More recently, in a 14-day, single-blinded crossover study with commercial diets supplemented with flaxseed mucilage topically, lower fat digestibility and looser stools were noted, compared with control-fed dogs [26]. In our study, we did observe an increase in total fecal output and metabolizable energy in the flax-containing diet, compared with control. Flaxseed was added at the expense of brown rice, oats, and barley, keeping chicken fat constant between the diets. These grains are relatively low in fat in comparison to flaxseed, making the flaxseed diet higher in fat, and may explain the greater ME content. Overall, addition of a moderately fermentable ingredient such as flaxseed resulted in no changes to total-tract nutrient digestibility or fecal consistency when fed to adult cats. Increased total fecal output may be attributed to the added insoluble and soluble fiber that whole ground flaxseed brings to the diet.

The main finding in this study was that feeding flaxseed at 2.6% inclusion resulted in continuing increases in serum ALA concentrations at 14 and 28 days of feeding, that overall were greater than in control-fed cats. Both diets did result in increases from day 0 in serum ALA due to chicken fat having low yet appreciable concentrations of the fatty acid (1–2% of total fatty acids). Most investigations on dietary n-3 PUFA bioavailability in cats are focused primarily on ingredients that directly supply DHA and EPA from algal and fish oils [29,30,31]. Ingredients that supply DHA and EPA n-3 PUFA are often the focus on contributing to the immunomodulatory effects rather than ALA due to cats having low delta-6 desaturase activity and lack of conversion of ALA to DPA and EPA. However, all three n-3 PUFAs play important roles in modulating the immune system through synthesis of prostaglandins and leukotrienes [32] and decrease inflammatory mediators released by T-cells [12]. Specifically, improving concentrations of ALA can improve skin moisture retention and result in skin and coat improvements in pets with poor coat quality [33]. 

Very few studies in cats have investigated bioavailability of ALA from dietary flaxseed and the potential anti-inflammatory effects in cats. However, a study in older cats fed a diet formulated to a 5:1 omega-6:omega-3 PUFA ratio from flaxseed oil (1.1% inclusion) increased serum ALA concentrations over 12 weeks of feeding [13]. Furthermore, cats consuming both flaxseed and fish oil diets exhibited suppression in inflammation by decreasing skin hypersensitivity to histamine and lowered B-cell, total T-cell, T-helper subset populations, and leukocyte proliferation response to pokeweed mitogen [13]. In our current study, dietary ALA concentrations were lower in comparison to Park et al. (0.5% vs. 4.0%), which explains why cats in our study had lower serum ALA concentrations [13]. However, both studies fed ALA at concentrations that exceed the NRC (2006) recommendation (0.05 g per 1000 kcal ME) [7]. 

Bioavailability of n-3 PUFA concentrations (i.e., ALA) in response to dietary flaxseed products (whole ground seeds and oil) has primarily been investigated in dogs. When dogs were fed a high-ALA flaxseed for 84 days, serum concentrations of ALA, DPA, and EPA increased after just 4 days of feeding [3]. When flaxseed oil was supplemented at 100 mL/kg of food to dogs for 3 weeks, serum ALA and EPA increased; however, breed differences were observed, where greyhounds exhibited greater bioavailability compared with beagles [4]. It was also found that ALA and LA increased from d15 to d22, suggesting that enrichment of these fatty acids in circulation is not stabilized after 3 weeks of feeding [4]. Interestingly, a more recent study in both beagles and greyhounds investigated feeding 100 mg/kg food of flaxseed oil on plasma fatty acid levels and their link to inflammatory responses such as heat shock protein 90, heat shock protein 70, and interleukin 1-beta [5]. There, flaxseed oil supplementation increased ALA and EPA concentrations and downregulated the expression of heat shock protein 90 and interleukin 1-beta in greyhounds but not in beagles. Both ALA and EPA were negatively correlated with these inflammatory markers in greyhounds. It is widely known that ALA can contribute to elongation for the synthesis of EPA, which has known immune benefits. However, at least in vitro, ALA has been reported to upregulate anti-inflammatory pathways through NF-κB and mitogen activated protein kinase [10,11]. It is also important to note that previous evidence in dogs has shown that serum n-3 PUFA concentrations continue to increase out to 90 days of feeding [34]. The current investigation in cats only went out to 28 days of feeding, suggesting that the ALA serum saturation point may be longer than the feeding duration. Overall, flaxseed and flaxseed oil feeding in both dogs and cats result in increases in ALA and EPA in dogs and ALA in cats [3,5,13]. Modulation of these fatty acids results in anti-inflammatory outcomes in both dogs and cats [5,13].

## 5. Conclusions

In conclusion, the current study indicates that addition of ground flaxseed to adult cat foods provides bioavailable ALA through 28 days of feeding, without negative impact on diet fatty acids, nutrient digestibility, or stool quality. This also indicates that ground flaxseed can contribute to omega-3:omega-6 PUFA balance by contributing to dietary n-3 PUFA. Some evidence exists in both dogs and cats that ALA from flaxseed oil may contribute to improved skin and serum inflammatory responses. These results support the inclusion of ground flaxseed in cat diets as a source of ALA. More studies to investigate (1) the benefits of adding ground flaxseed and ALA’s contribution to skin and coat health and inflammation management and (2) the time at which serum ALA concentrations stabilize in cats are needed.

## Figures and Tables

**Figure 1 animals-12-02543-f001:**
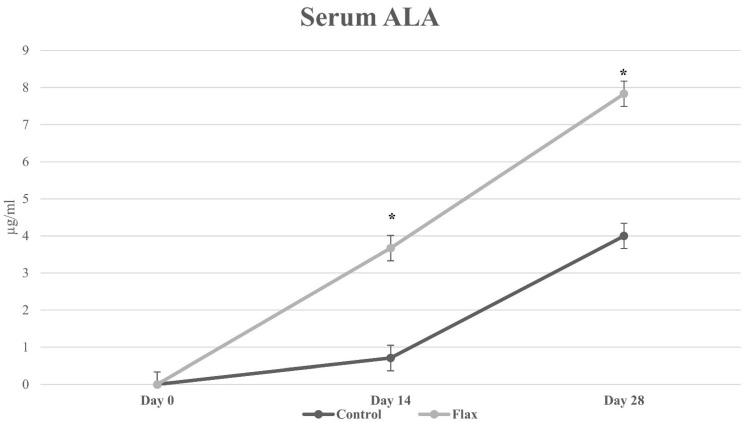
Average serum alpha-linolenic acid (ALA) concentrations in adult cats fed control and 2.6% as-is whole ground flaxseed diets. * Denotes a significant (*p* < 0.05) difference between diets.

**Table 1 animals-12-02543-t001:** Ingredient and chemical composition of experimental diets (Control and Flax).

Ingredient	Amount	
	%, as-is	
	Control	Flax
Mechanically Deboned Chicken	18.35	18.35
Chicken Meal	18.16	18.16
Brown Rice	14.48	13.61
Oats	7.80	6.92
Barley	7.71	6.84
Pea Protein	7.04	7.04
Chicken Fat	5.82	5.82
Dried Egg	5.24	5.24
Peas	4.37	4.37
Tomato Pomace	3.02	3.02
Flaxseed	-	2.62
Fish meal	1.99	1.99
Powdered Digest	1.75	1.75
Low inclusion ingredients *	4.06	4.06
Analyzed Composition		
Dry Matter (DM), %	94	96
Metabolizable Energy, kcal/g	3.87	4.04
Organic Matter, %	94	91.9
Crude Fat, %	19.3	20.9
Crude Protein, %	35.1	35.8
Crude Fiber, %	2.80	2.50
ALA, mg/g DM	1.44	4.77
Total Fatty Acids, mg/g DM	183	179

* Consists of ingredients added at <0.62% including calcium chloride, calcium sulfate, choline chloride, methionine, salt, potassium chloride, taurine, vitamin and mineral premix, inulin, alfalfa meal, mixed tocopherols, sweet potato, carrots, and cranberry.

**Table 2 animals-12-02543-t002:** Average weekly food intake, final body weight, and total-tract apparent nutrient and ALA digestibility in adult cats fed control and 2.6% as-is whole ground flaxseed diets.

	Dietary Treatments			
Item	Control	Flax	SEM	*p*-Value
Food Intake, g/day	62.9	66.0	1.81	0.24
Final BW, kg	4.00	3.80	0.27	1.00
Digestibility, %				
Dry matter	80.6	80.3	1.03	0.85
Crude Protein	82.8	82.4	0.97	0.74
Crude fat	91.5	91.1	0.63	0.64
Energy	88.1	87.8	0.63	0.70
ALA	97.5	97.8	0.53	0.67
Metabolizable Energy, kcal/g	3.90	4.20	0.04	*< 0.01*

All values are means and pooled SEM. Italicized *p*-values denote a significant (*p* < 0.05) difference between diets.

**Table 3 animals-12-02543-t003:** Fecal score, total fecal output, fecal moisture, total fecal fatty acids, and fecal ALA in adult cats fed control and 2.6% as-is whole ground flaxseed diets.

	Dietary Treatments			
Item	Control	Flax	SEM	*p*-Value
Fecal score	2.92	2.68	0.20	0.24
Fecal output, g	185	229	14.1	*0.04*
Fecal moisture, %	69.1	72.3	2.02	0.17
Total Fecal Fatty Acids, mg/g	41.9	49.2	3.40	0.09
Fecal ALA, mg/g	0.19	0.52	0.12	0.82

All values are means and pooled SEM. Italicized *p*-values denote a significant (*p* < 0.05) difference between diets.

**Table 4 animals-12-02543-t004:** Hematology and serum chemistry results of adult cats fed control and 2.6% as-is ground flaxseed diets.

	Dietary Treatments								
	Control	Flax	Control	Flax	Control	Flax	*p*-Value		
Item	Day 0		Day 14		Day 28		Diet	Time	Diet × Time
Total Protein, g/dL	7.0 ± 0.15	6.9 ± 0.17	6.8 ± 0.13	6.8 ± 0.16	6.9 ± 0.16	7.1 ± 0.19	0.98	0.40	0.79
Albumin, g/dL	3.1 ± 0.06	3.1 ± 0.07	3.1 ± 0.06	3.1 ± 0.08	3.2 ± 0.05	3.2 ± 0.09	0.91	0.43	0.77
Globulin, g/dL	3.9 ± 0.16	3.8 ± 0.19	3.7 ± 0.16	3.7 ± 0.18	3.8 ± 0.19	3.9 ± 0.23	0.95	0.70	0.77
A:G Ratio	0.8 ± 0.04	0.8 ± 0.06	0.9 ± 0.05	0.8 ± 0.06	0.9 ± 0.05	0.9 ± 0.07	0.94	0.80	0.90
AST, U/L	24 ± 1.13	24 ± 1.53	32 ± 6.49	26 ± 2.49	19 ± 0.88	20 ± 1.38	0.48	*0.01*	0.50
ALT, U/L	45 ± 2.22	52 ± 3.31	49 ± 2.63	53 ± 3.86	49 ± 3.39	50 ± 2.74	0.11	0.68	0.66
Alkaline Phosphatase, U/L	24 ± 3.10	25 ± 3.24	25 ± 2.60	23 ± 2.93	24 ± 2.37	24 ± 2.97	0.94	0.98	0.89
GGTP, U/L	2.0 ± 0.25	2.0 ± 0.17	2.0 ± 0.33	2.0 ± 0.21	1.0 ± 0.10	1.0 ± 0.15	0.45	0.01	0.42
Total Bilirubin, mg/dL	0.1 ± 0.0	0.1 ± 0.0	0.1 ± 0.0	0.1 ± 0.0	0.1 ± 0.0	0.1 ± 0.0	1.00	1.00	1.00
Urea Nitrogen, mg/dL	26 ± 0.98	25 ± 1.13	23 ± 0.93	24 ± 1.27	24 ± 1.27	23 ± 0.92	0.50	0.04	0.82
Creatinine, mg/dL	1.3 ± 0.08	1.4 ± 0.07	1.4 ± 0.09	1.5 ± 0.08	1.6 ± 0.08	1.5 ± 0.07	0.48	0.03	0.64
BUN/Creatinine Ratio	22 ± 1.58	19 ± 0.67	17 ± 0.84	17 ± 1.09	15 ± 0.65	15 ± 0.82	0.14	*<0.01*	0.21
Phosphorus, mg/dL	3.7 ± 0.25	3.9 ± 0.18	3.8 ± 0.16	4.2 ± 0.23	4.5 ± 0.21	4.8 ± 0.16	0.06	0.01	0.92
Glucose, mg/dL	95 ± 7.63	87 ± 3.16	86 ± 3.66	93 ± 5.93	98 ± 7.33	92 ± 2.59	0.60	0.64	0.36
Calcium, mg/dL	8.9 ± 0.10	9.0 ± 0.10	8.9 ± 0.09	9.0 ± 0.10	9.1 ± 0.09	9.1 ± 0.09	0.37	0.26	0.73
Magnesium, mEq/L	1.6 ± 0.04	1.7 ± 0.04	1.6 ± 0.04	1.6 ± 0.04	1.7 ± 0.03	1.6 ± 0.04	0.67	0.11	0.23
Sodium, mEq/L	150 ± 0.37	150 ± 0.42	149 ± 0.94	150 ± 0.60	150 ± 0.50	150 ± 0.45	0.83	0.78	0.33
Potassium, mEq/L	4.2 ± 0.11	4.5 ± 0.12	4.3 ± 0.11	4.5 ± 0.10	4.4 ± 0.09	4.5 ± 0.08	*0.02*	0.96	0.53
Chloride, mEq/L	119 ± 0.55	119 ± 0.28	118 ± 0.83	120 ± 0.64	119 ± 0.41	119 ± 0.33	0.37	0.99	0.38
Cholesterol, mg/dL	127 ± 11.3	120 ± 5.51	130 ± 9.16	125 ± 6.99	141 ± 9.89	138 ± 8.93	0.49	0.18	0.98
Triglycerides, mg/dL	31 ± 3.06	28 ± 2.11	25 ± 1.77	24 ± 1.56	25 ± 1.63	26 ± 1.33	0.48	0.03	0.54
CPK, U/L	320 ± 146	283 ± 54.8	410 ± 213	286 ± 65.9	151 ± 11.6	171 ± 18.1	0.61	0.22	0.81
WBC,10^3^/mm^3^	11.0 ± 0.61	11.3 ± 0.90	9.9 ± 1.04	10.7 ± 0.47	10.1 ± 0.61	11.3 ± 0.78	0.22	0.56	0.82
RBC, 10^6^/mm^3^	7.6 ± 0.25	7.8 ± 0.26	7.6 ± 0.24	7.4 ± 0.31	7.9 ± 0.28	7.8 ± 0.25	0.98	0.43	0.81
Hemoglobin, g/dL	10.9 ± 0.35	11.0 ± 0.31	10.5 ± 0.40	10.2 ± 0.35	10.6 ± 0.30	10.4 ± 0.33	0.71	0.15	0.81
Hematocrit, %	36 ± 1.05	37 ± 1.12	33 ± 1.03	33 ± 1.14	36 ± 1.13	36 ± 1.04	0.57	*0.01*	0.75
MCV, um^3^	48 ± 0.79	48 ± 0.81	44 ± 0.87	44 ± 0.70	46 ± 0.90	46 ± 0.96	0.41	*0.01*	1.00
MCH, uug	14.4 ± 0.31	14.2 ± 0.23	13.9 ± 0.36	13.8 ± 0.27	13.5 ± 0.34	13.3 ± 0.25	0.51	*0.01*	0.99
MCHC, g/dl	30.4 ± 0.30	29.6 ± 0.25	31.7 ± 0.35	31.1 ± 0.24	29.7 ± 0.38	28.9 ± 0.22	*0.01*	*<0.01*	0.97
Platelets, 10^3^/mm^3^	349 ± 23.2	384 ± 24.3	293 ± 23.0	356 ± 21.0	318 ± 20.7	367 ± 20.6	*0.01*	0.17	0.82
Absolute Polys	6486 ± 527	6435 ± 641	5830 ± 997	6433 ± 570	5535 ± 422	6891 ± 634	0.24	0.87	0.57
% Polys	59 ± 3.34	58 ± 4.08	57 ± 4.11	60 ± 4.56	55 ± 3.27	61 ± 3.23	0.41	0.99	0.65
Absolute Lymphoctyes	3309 ± 405	3512 ± 598	3047 ± 383	3096 ± 571	3433 ± 350	3100 ± 493	0.95	0.78	0.85
% Lymphocytes	30 ± 3.01	31 ± 4.28	33 ± 3.74	29 ± 4.67	34 ± 2.76	28 ± 3.96	0.30	0.99	0.65
Absolute Monocytes	335 ± 54.9	298 ± 45.6	239 ± 50.2	223 ± 26.7	245 ± 50.9	219 ± 47.8	0.49	0.12	0.97
% Monocytes	3.0 ± 0.37	3.0 ± 0.27	2.0 ± 0.40	2.0 ± 0.23	2.0 ± 0.37	2.0 ± 0.35	0.19	0.10	0.99
Absolute Eosinophils	851 ± 62.4	1004 ± 104	773 ± 72.8	938 ± 115	837 ± 94.7	1060 ± 166	0.05	0.67	0.94
% Eosinophils	8.0 ± 0.79	9.0 ± 0.70	8.0 ± 0.49	9.0 ± 1.38	8.0 ± 0.78	9.0 ± 1.19	0.18	0.94	1.00

All values are means ± SEM. Italicized *p*-values denote a significant (*p* < 0.05) difference for that effect.

## Data Availability

The data presented are available in this manuscript.

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
