# Peer review of "Dietary Ground Flaxseed Increases Serum Alpha-Linolenic Acid Concentrations in Adult Cats"

_animals, 2022, doi:10.3390/ani12192543_

Round 1

Reviewer 1 Report

Lines 13, 16, 231, 244, 253 – the term ‘detriment’ has been used frequently, maybe it could be more diversified?

Line 15 – stool ‘consistency’

Line 16 – replace ‘feeding flaxseed’ with ‘flaxseed supplemented diet’

Line 18 – ‘…modulate SERUM fatty acid concentrations…’

Line 77 – ‘ALA HAS BEEN demonstrated…’

Line 89 – ‘…may be misunderstood to mean…’ I suggest rephrasing towards clarification

Section 2.1. – could you please explain if there was a dietary transition period? What type of food all experimental animals were fed? why the cats were fed once per day? Were there no (complete) refusals? The diets/formulation were extruded? You did not expect thermal degradation (of the lipid fraction) during processing?

Lines 198, 200 – I’d suggest adding ‘flaxseed supplemented diet’

Reviewer 2 Report

The study evaluated the effects of dietary ground flaxseed on fecal and serum alpha-linolenic acid concentrations, nutrient digestibility, and stool quality in cats. It is well written and brings new data to be used in cat nutrition.

General comments:

Define the acronyms at the first use and use them consistently (e.g.: DM, CP, ME…)

Specific comments:

L101: do not put parenthesis inside the parenthesis

L187, Table 2. Food intake is g/animal/day and not /week, right?

L203. Table 4. If the data is normally distributed (it is presented as means), why not including a pooled SEM than each SEM?

L208-210. But in this case, the studies found those effects with specific sources of EPA and DHA and not ALA.

L278: Change to NRC (2006)…
